

# A single framework for assessing flash flood and landslide susceptibility: an application to the Mediterranean

Alessia Riveros[1], Chamidu Gunaratne[2], Mario Martinelli[3,4], Frederiek C. Sperna Weiland[1]

[1]Department of Catchment and Urban Hydrology, Deltares, P.O. Box 177, 2600 MH Delft, The Netherlands
[2]Department of Safe and Resilient Infrastructure, Deltares, P.O. Box 177, 2600 MH Delft, The Netherlands
[3]Department of Soil, Water and Structures, Deltares, P.O. Box 177, 2600 MH Delft, The Netherlands
[4]Department of Civil and Environmental Engineering, Carleton University, K1S 5B6 Ottawa, Canada

*Correspondence to*: Alessia Riveros (Alessia.Riveros@deltares.nl)

**Abstract.** Flash floods and landslides have caused severe economic damages and loss of life, especially in mountainous regions. To support effective risk management there is a growing interest in multi-hazard assessment. In this study a globally applicable Machine Learning (ML) Framework for landslide and flash flood susceptibility mapping was applied and evaluated in the Italian region Liguria that is frequently and severely impacted by both hazards. A relatively dense inventory of past events was constructed to facilitate the training of the ML Framework. The analysis revealed substantial similarities in the
causative factors for the two hazards. There is a considerable area of Liguria susceptible to both hazards, although flash floods most often occur in river valleys whereas landslide susceptibility is also high in the upper courses of river catchments. We found a very high susceptibility along the coastline where many villages and cities are located. The unified framework allows for the integration of different hazard types under a consistent modelling structure. This enhances the comparability of results and supports the development of integrated mitigation strategies for any region of interest.

## 1 Introduction

Both flash floods and landslides have in the past led to severe economic damages and loss of life, especially in mountainous regions (Anon, 2024; Fedato et al., 2023; Gaume et al., 2009; Wood et al., 2016). There are indications that climate change, and the related changes in extreme rainfall events, will cause an increase in their intensity and frequency of occurrence (Terzi et al., 2019; Wood et al., 2016; Zander et al., 2022). Changes in socio-economic conditions, such as deforestation, land-use
changes and occupation of flood prone areas, will likely contribute to further increases in economic loss (Hurtado-Pidal et al., 2022; Llasat, 2021; Muñoz-Torrero Manchado et al., 2022).

Landslides and flash floods share similar (pre-)conditions of occurrence, i.e. steep mountainous terrain, deforested and highly erodible slopes, low permeability, saturated soils and extreme rainfall as triggering factor (Borga et al., 2014; Terranova and Gariano, 2014). It is widely recognised that the impacts of multi-hazard events can surmount those of multiple single hazards
(Gill and Malamud, 2014; Hochrainer-Stigler et al., 2023; UN. Secretary-General and UN. Open-ended Intergovernmental Expert Working Group on Indicators and Terminology relating to Disaster Risk Reduction, 2016; Zscheischler et al., 2018).



As a consequence, there is a growing focus on multi-hazards within risk assessment to support improved risk management and adaptation planning (Adnan et al., 2025; Hochrainer-Stigler et al., 2023; Schlumberger et al., 2022).

A crucial step towards multi-hazard risk assessment is susceptibility mapping. Susceptibility maps help to identify regions that are more likely to face a given hazard and provide important input for disaster risk management and adaptation planning. The susceptibility presents the likelihood of the occurrence of an event in an area based on the local terrain conditions (Tiggeloven et al., 2025; Wilde et al., 2018). Susceptibility maps are typically generated with statistical methods or Machine Learning (ML) algorithms trained on a set of past events (Alarifi et al., 2022; Bui et al., 2019; Chowdhury, 2024; Elghouat et al., 2024; He et al., 2025; Khodaei et al., 2025; Luu et al., 2023; Pham et al., 2021; Shahabi et al., 2021; Wahba et al., 2024). In line with these methods, Tehrani at al. (2021) developed a globally applicable framework for landslide susceptibility. The framework builds upon global datasets of amongst others; terrain, slope, land cover, vegetation and river networks.

In this follow up study, we transferred the susceptibility framework of Tehrani et al. (2021) into a framework for landslide and flash flood susceptibility mapping. This study contributes by applying the unified susceptibility framework to a region with contrasting geomorphological and climatic conditions, focussing on static conditioning factors. The sensitivity of the two hazards towards a similar set of driving factors is evaluated and the differences in resulting landslide versus flash flood susceptibility maps is analysed. Liguria, located in north-western Italy, provides an ideal case study due to its high susceptibility to both flash floods and landslides, driven by steep terrain, short hydrological response times, and frequent high-intensity rainfall.

Although the framework is developed around global datasets, it requires local inventories of past hazard events to train the underlying machine learning algorithm. Due to the limited spatial and temporal extent and the relative remoteness of occurrence of both hazards such inventories are often incomplete or limited and thus influence the quality of the resulting susceptibility maps (Modrick and Georgakakos, 2015). For the relatively data rich region of Liguria we have access to three open archives of past landslide and flash flood events.

With this study, we specifically aimed to (1) evaluate the similarity in the causative factors for flash floods and landslides, (2) apply a susceptibility mapping algorithm consistently to both hazards, and (3) assess the accuracy of the resulting susceptibility maps.

## 2 Study Area

Liguria is a region located in the Northwest of Italy (see Figure 1) covering an area of 5410 km$^2$. It is bordered by the Ligurian Sea to the South, by the Maritime Alps to the Northwest, and by the Apennines to the Northeast. Characterized by a Mediterranean climate, summers in Liguria are driest whereas falls are wettest. Fall exhibits a positive trend in terms of cumulative precipitation from the period of 1981-2010 with respect to 1961-1990 (Giacomo Agrillo and Veronica Bonati, 2013). Annual precipitation varies from less than 600 mm with 50 rainy days per year on the west coast to more than 2000 mm and around 100 rainy days per year in the Apennines (Giacomo Agrillo and Veronica Bonati, 2013).





Due to Liguria's geographical location, it is characterized by steep slopes and mountainous terrain reaching a maximum of
2201 m.a.s.l. The lithology is constituted mainly by claystones and clays (55%) which has a tendency to split into thin, flat
layers with low porosity, and by schist, phyllites, quartzites and marbles (14%). The dominant land covers are closed forest
(76%) followed by cropland (14%).

The hydrology in Liguria is characterized by medium size catchments ranging from 10 to 1000 km² (Silvestro et al., 2018) and
short response times ranging from 1 to 5 hours.

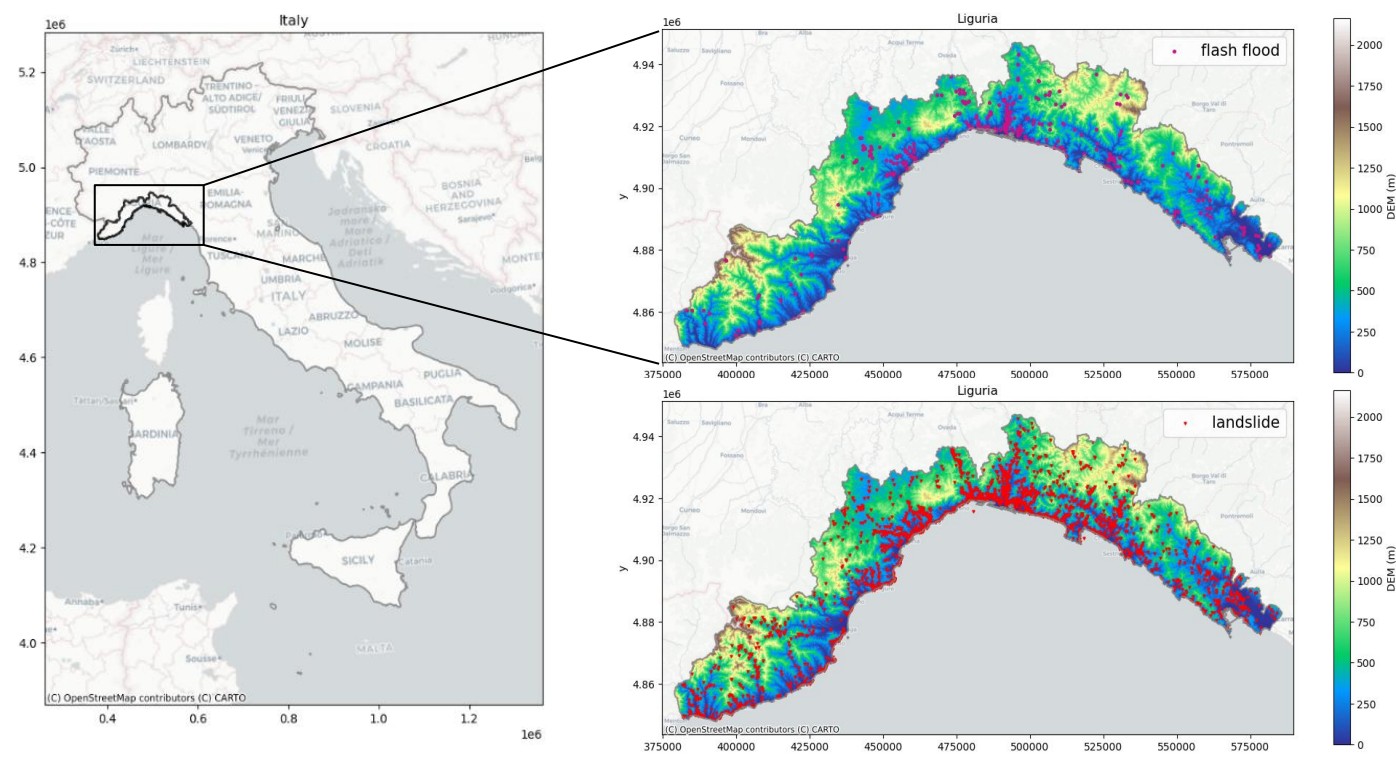

**Figure 1: Map showing the location of the Liguria region in Italy (left). Top-right (bottom-right) shows the overview of past flash flood (landslides) events in Liguria.**

## 3 Data and Methods

### 3.1 Data

#### 3.1.1 Flash Flood and Landslide Inventories

**Flash Floods**

In this study we define flash floods as high intensity short duration floods caused by storm durations of up to 48 hours following
the definition of Amponsah et al., (2018) on flash floods in Mediterranean regions. Other Mediterranean flash floods are



defined as a short duration of less than 1 hour to 24 hours (Gaume et al., 2009) but given the uncertainty in the inventory, it was deemed too restrictive. Only flash floods caused by hydrometeorological conditions or landslides are considered, those caused by infrastructure failure (e.g., dam breaches) are excluded. Historical flash floods have been retrieved from two sources: (1) the AVI (*Aree Vulnerate in Italia*), a database developed by the National Group for the Prevention of Hydrogeologic Hazards of the Italian National Research Council (CNR) which includes the ID, event type, date, river name, notes,

hydrological data, cause and coordinates of floods recorded throughout Italy from 1951-1994 and is comprised of 256 flash floods points after processing.

In case of duplicate flash flood entries, where records shared the same point geometry and date but had different river names, one entry was retained unchanged and an additional point was added with updated coordinates located on the nearest point on the corresponding river, using river and stream data from Open Street Map.

(2) ARPAL (*Agenzia  regionale per la protezione dell'ambiente ligure*) provides a collection of PDF reports of significant meteorological and hydrological events by date in Liguria including point rainfall information, effects on the ground, and relevant damages from 2009 to 2024 leading to 282 flash flood points after processing.

We used a Large Language Model (LLM) – GPT-4O via the Azure OpenAI Service (Microsoft, 2025) to process ARPAL event reports summarizing past geohydrological hazards in the Liguria region, automatically extracting event information and

generating structured outputs based on our predefined database schema. For the automated extraction of flash floods we framed the search corresponding to floods, as the exact term 'flash flood' is never used in the reports. From these PDF documents, specific event details - including point rainfall amount, duration, flood type and coordinates – could automatically be extracted. Subsequently, we verified that each description corresponded to flash flood and removed the ones that didn't.

The resulting flash flood events are displayed in Figure 1 (top-right).


**Landslides**

In this study only rainfall induced landslides are considered. Similarly to flash floods we used the (i) AVI database which included 620 landslides events along with their coordinates, id, cause, notes and date, and the (ii) ARPAL reports resulting in 97 landslide events. Both provide events at daily resolution. Additionally, we used the ITALICA (Italian rainfall-induced

LandslIdes Catalogue) (Peruccacci et al., 2023) inventory with landslide id, coordinates and dates covering the period from 2002 to 2021. It comprises 1,699 landslide events in Liguria, documented at hourly temporal resolution. As the landscape has altered over time due to human influences we decided to only include events reported after 1940. The resulting landslide events are displayed in Figure 1 (bottom-right).

Figure 2 displays the distribution of flash flood (left) and landslide events (right) over the calendar months. For both hazards

most events occurred in fall (September, October, November). Flash flood occurrence is highest in September, most likely because soils are dry and do not infiltrate well after a long dry summer season. Landslides are more spread out throughout the year.



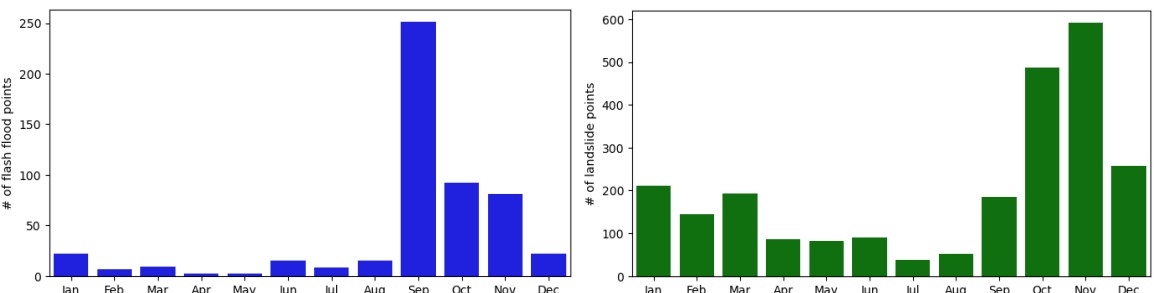

**Figure 2: Number of flash flood (left) and landslide (right) events in the created inventory sorted by calendar month.**

### 3.1.2 Causative factors for flash floods and landslides

Based on a literature review, an initial set of landslides and flash flood causative factors was identified (Alarifi et al., 2022; Bui et al., 2019; Chowdhury, 2024; Elghouat et al., 2024; He et al., 2025; Khodaei et al., 2025; Luu et al., 2023; Pham et al., 2021; Rayamajhi et al., 2025; Shahabi et al., 2021; Wahba et al., 2024) These factors are listed in Table 1 and briefly described afterwards. Figure 3 and Figure 4 provide a visual overview of the spatial distribution of these input layers, as used in the machine learning algorithm.

Unlike other studies that include rainfall as a dynamic input variable in landslide susceptibility mapping (Ahmed et al., 2023; Lee et al., 2022), we chose not to incorporate rainfall data in our model. Although rainfall is a well-known trigger for both landslides and flash floods, its use as a predictive requires rainfall datasets with sufficient spatial and temporal resolution. This is particularly important in mountainous areas such as Liguria, where rainfall patterns are highly variable, and short-duration, high-intensity convective storms frequently occur. These are often not captured by coarse-resolution datasets, such as those provided by satellite-based or global reanalysis products. Moreover, the other input data used in this study - such as topography and land cover - are available at much finer resolutions. Such a mismatch between input resolutions can lead to inaccuracies in the model and reduce confidence in the resulting susceptibility maps. Future studies may incorporate dynamic rainfall information, provided that high-resolution precipitation datasets become available and are spatially consistent with the other model inputs.

**Table 1: Overview of causative factors for landslides and flash floods and the data sources used.**

| Factor | Source | Spatial resolution |
|--------|--------|--------------------|
| DEM | Eurostat (European Commission - DG ENTR, 2012) | 1 arc per second or ~30 m |
| Slope | Derived from DEM | 1 arc per second or ~30 m |
| Aspect | Eurostat (European Commission - DG ENTR, 2012) | 1 arc per second or ~30 m |
| Curvature | Derived from DEM | 1 arc per second or ~30 m |





| TWI | Derived from DEM and slope | 1 arc per second or ~30 m |
|---|---|---|
| SPI | Derived from DEM and slope | 1 arc per second or ~30 m |
| Distance to rivers | Derived from Open Street Map (Open Street Map contributors, n.d.) | There is no consistent spatial resolution |
| Land cover | JRC (Joint Research Centre (JRC), European Commission, 2018) derived from the Global Land Cover Map for 2009 | 200 m |
| NDVI | NASA MODIS (NASA LP DAAC, 2017) | 250 m |
| Lithology | JRC (Joint Research Centre (JRC), European Commission, 2018) derived from the International hydrogeological map of Europe 1:1,500,000 (IHME1500) (2019) | 200 m |
| Distance to roads | Derived from Open Street Map, (Open Street Map contributors, n.d.) | There is no consistent spatial resolution |



**Figure 3: Overview of landslide and flash flood numerical causative factor maps used as input for the ML algorithms.**

135





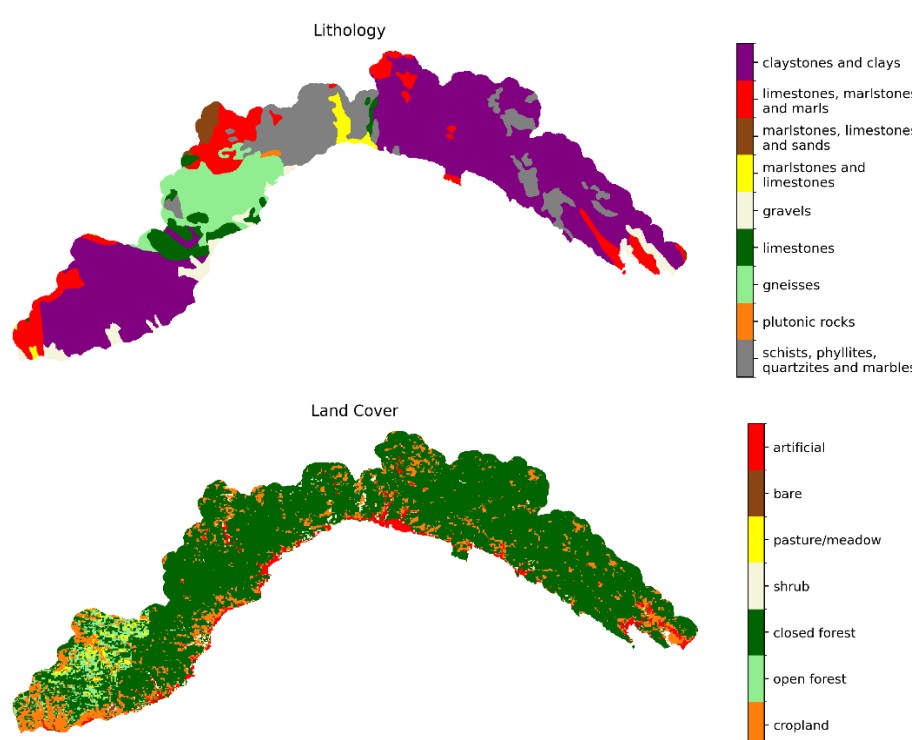

**Figure 4: Overview of landslide and flash flood categorical causative factor maps used as input for the ML algorithms.**

**Topographical factors**

The Digital Elevation Model (DEM), slope, aspect, and curvature play an important role in the velocity and direction of surface runoff. They are related to the stability of slopes affecting the occurrence of landslide events and to the presence of valleys where floods are most likely to occur.

**Hydrological factors**

The Topographic Wetness Index (TWI), the Stream Power Index (SPI), and the proximity to river are proxies of the hydrological conditions of the landscape. The TWI is a proxy for determining the susceptibility of areas to (soil) saturation in hilly areas. It is defined in Beven & Kirby (1979) as:

$$TWI = \ln \left( \frac{A}{\tan (\beta)} \right) \tag{1}$$

where $A$ is the drainage area per unit contour length and represents the water flow accumulation at a point i, and $\beta$ is the local slope angle in radians at the same point. The SPI measures the erosive power of flowing water that increases with the steepness of the slope and the larger upstream accumulation area for water and is defined as:

$$SPI = A * \tan (\beta) \tag{2}$$



**Landscape factors**

Land cover and the Normalized Difference Vegetation Index (NDVI), which quantifies the amount and health of vegetation, influence the infiltration capacity and stability of the topsoil, as well as the runoff speed.

**Geologic factor**

The sub surface lithology influences the water storage and the proneness or resistance to sliding movements.

**Anthropogenic factor**

The proximity to roads interferes with the slope stability and can also capture the bias to the recording of landslide events.

## 3.2 Methods

### 3.2.1 Multicollinearity and frequency ratio analysis

**Variance Inflation Factor**

In order to run the hazard susceptibility modelling, we must make sure the input variables (factors) are not collinear. High collinearity between factors decreases the model's interpretability and accuracy. Hence, we used the Variance Inflation Factor (VIF) to quantify the collinearity of the input variables.

The VIF is a factor by which the correlations among the predictors or independent variables increase the variance:

$$VIF_i = \frac{1}{1-R_i^2} \tag{3}$$

Where $R_i^2$ is the coefficient of determination for regressing the $i^{th}$ independent variable on all the other predictors. Hence, if the predictors are uncorrelated $R_i^2$ is equal to 0 and VIF equals one. VIF values between 1 and 5 indicate a low to moderate level of multicollinearity, values between 5 and 10 correspond to a moderate to high level of correlation, whereas values higher than 10 show a strong multicollinearity.

**Frequency Ratio**

To analyse the conditions that led to the occurrence of both past flash floods and past landslides, we used the frequency ratio. The frequency ratio is a method used to quantify the relationship between the spatial distribution of past hazard occurrences (e.g., flash floods or landslides) and individual classes of a causative factor. It is calculated as the ratio between the proportion of hazard pixels in a given class and the proportion of total area covered by that class.

$$FRv_i = \frac{Fc_i/Fs}{Ac_i/As} \tag{4}$$

where $Fc_i$ is the number of pixels with flash floods (or landslides) for each class of each $i$ variable, $Fs$ the total number of pixels with flash floods (or landslides) in the study area, $Ac_i$ the number of pixels for each class of each $i$ variable, and $As$ the total number of pixels in the study area.



A frequency ratio greater than 1 indicates a positive association between the class and the hazard occurrence (i.e., the hazard is more likely in that class), while a ratio less than 1 suggests a negative association. This method was applied separately for both landslides and flash floods, enabling comparison of how each hazard correlates with different classes of conditioning factors.

For continuous variables, we used the Jenks natural breaks classification (George F. Jenks, 1967) to divide each variable into
four classes sampling randomly 10% of the data. This method minimizes intra-class variance and maximizes inter-class variance. Only for aspect and curvature, a manual classification based on geomorphological properties was used to better support the interpretation of the results.

### 3.2.2 Hazard Susceptibility Modelling

The hazard susceptibility modelling framework  (see Figure 5) is a GIS tool programmed in Python that creates landslide and flash flood susceptibility maps using three different Machine Learning (ML) Models Logistic Regression (LR), Random Forest (RF), Support Vector Machine (SVM). In this study the input and training data was collected for both landslide and flash flood occurrences to allow for a consistent susceptibility assessment for both flash floods and landslides.

Logistic Regression (Cox, 1958) is a supervised machine learning algorithm especially used for binary classification problems.
It uses a sigmoid function to predict the probability the input data belongs to one of the two classes.

The Random Forest (Breiman, 2001) is the most used machine learning algorithm, as it is a popular ensemble learning method that is widely used for classification and regression. It is a combined model that integrates multiple decision trees. The Random Forest involves two concepts: 1) random sampling of data points; 2) segmentation of nodes based on feature subsets. Each tree is trained on a sample of data points drawn at random, and these samples are drawn repeatedly. At each node, the decision tree
will consider segmentation based on a part of the feature. The results from all decision trees are aggregated and the result of the Random Forest is obtained.

Support Vector Machine (Cortes and Vapnik, 1995) is a supervised machine learning algorithm used for classification and regression analysis. It is suitable for binary classification where it aims to find the best decision line or hyperplane separating the two classes. This is found when the distance or margin from the closest data points (support vectors) to the hyperplane is
the largest. In case the data cannot be separated linearly, the input data is mapped into a higher dimensional space in which a boundary can be more easily obtained.

The framework is capable of taking in both user-defined inputs - for example, high-resolution LiDAR data - as well as keeping the option to rely on global and/or publicly available assets. All inputs are reprojected to a reference one defined by the user which in our case was the DEM (1 arc per second).
The ML framework requires two types of input datasets. The first is the inventory of historical hazard occurrences - landslides or flash floods – provided in the form of spatial coordinates and timing. To account for the uncertainty in the exact location, is



the framework creates a 3x3 matrix around each point – this is definable by the user. This increases the number of positive samples relative to the original inventory.

Although there is no universally accepted rule for dividing the dataset into training and testing subsets (Joseph, 2022), ratios
of 80:20 or 70:30 ratio are commonly applied. In this study, the inventory was split using a 80:20 ratio to train and test the ML algorithm, respectively. The completeness of this inventory - both in time and space - is a main determining factor of the accuracy of the resulting landslide susceptibility map.

The second type of input type includes the causative and triggering factors influencing hazard occurrence (see Sect. 3.1.2 Causative factors for flash floods and landslides). These include topographic, hydrological, geological, landscape, and
anthropogenic variables that collectively define the susceptibility conditions across the study area.

The modelling workflow involves three main steps: model calibration, evaluation of the model's discriminatory power, and overall model performance assessment.

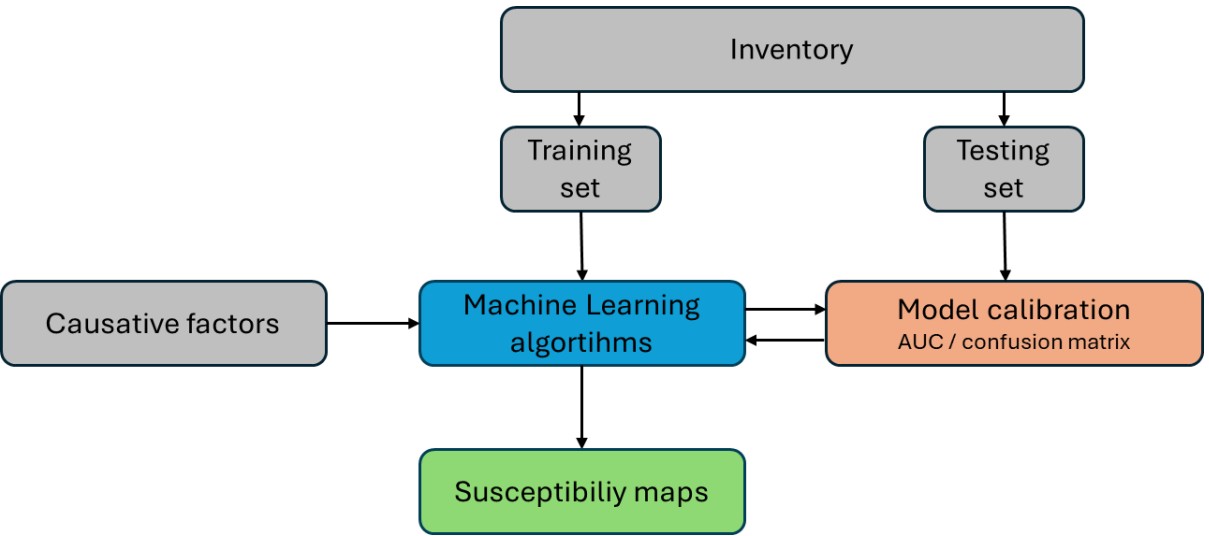

**Figure 5: The workflow of the ML based susceptibility mapping framework**

**Model calibration**

Prior to performance evaluation, the ML algorithm is calibrated to identify the optimal set of hyperparameters. The grid search method was employed to systematically test all possible parameter combinations and determine those yielding the highest predictive accuracy, as measured by the Area Under the Receiver Operating Characteristic (ROC) Curve (AUC). This ensures
that hyperparameter selection is guided by the ability of the model to distinguish between hazard and non-hazard conditions, rather than a single-threshold accuracy score.

This process was complemented by a five-fold cross-validation resampling technique in which the data was split into five folds maintaining the same proportion of occurrences and non-occurrences within each fold. In each iteration, four folds were used



for training and one for validation, cycling through all combinations to obtain an averaged performance score. The best-performing parameter configuration was then used to train the final models applied in subsequent analyses.

**Evaluating the discriminatory power of the framework**

After calibration, the models were evaluated for their ability to distinguish between the hazard and non-hazard conditions. The Receiver Operating Characteristic (ROC) and the Area Under the curve (AUC) metrics were used to evaluate the discriminatory power of the models.

The ROC curve represents a graphical plot of the model performance across all classification thresholds, showing the trade-off between the True Positive Rate (TPR) and False Positive Rate (FPR) at each threshold level. The AUC provides a single numerical measure summarizing this discriminatory power, i.e. the capacity to correctly distinguish landslide (or flash flood) from no-landslide (or flash flood) conditions.

An AUC value of 0.5 indicates random classification performance, while an AUC approaching 1.0 represents a model with excellent predictive capability (Uwihirwe et al., 2022).

**Model performance evaluation**

The predictive accuracy of each ML model is evaluated based on the AUC, the confusion matrix, and the accuracy. The confusion matrix records the number of True Positives (TP, i.e., the number of times the event occurrence is correctly predicted), True Negatives (TN, i.e., the amount of correctly predicted non-occurrences), False Positives (FP, i.e., the model predicts an occurrence when there is none), and False Negatives (FN, i.e., the model does not predict an occurrence when there is an event occurrence).

The accuracy of the model summarizes the proportion of correctly classified landslides (or flash floods) over the total number of landslides (or flash floods) and is calculated as:

$$Accuracy = \frac{\sum TP + TN}{\sum TP + FP + FN + TN} \tag{4}$$





# 4 Results

## 4.1 Analysis of the multicollinearity of causative factors

265    Figure 6 shows the calculated Variance Inflation Factor (VIF) for all selected causative factors. Only the Stream Power Index (SPI) exhibited moderate multicollinearity (VIF = 5.7), while all other factors showed low level of multicollinearity. Therefore, all factors were included in the ML-models.

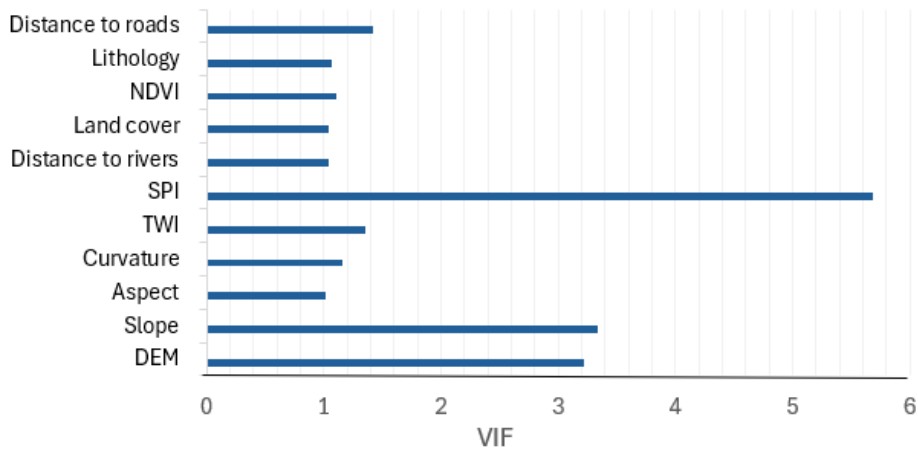

**Figure 6: Variance Inflation Factor (VIF) for the selected hazard causative factors.**

## 270    4.2 Frequency Ratio Analysis

Figure 7 and Figure 8 present the frequency ratio (FR) values for landslides and flash floods across numerical variables. Overall, the FR patterns were similar for both hazards.



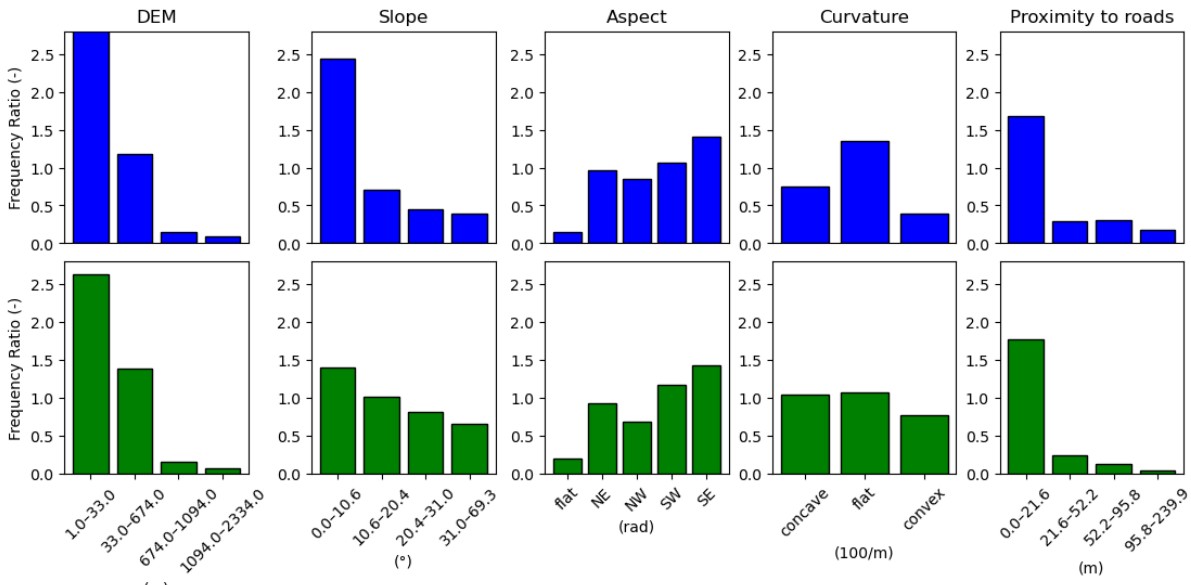

**Figure 7: Frequency ratio (FR) for flash floods (top) and landslides (bottom) for numerical variables**

Figure 7 shows that lower elevations and gentler slopes corresponded to higher likelihood of both flash floods and landslides occurrences. This observation is in line with previous studies on flash floods (Chowdhury, 2024; Elghouat et al., 2024; Pham et al., 2021; Rayamajhi et al., 2025). Although steep slopes tend to rapidly collect surface runoff, the water is accumulated in valleys.

For landslides, however, the apparent preference for gentle slopes is unexpected. The differences in FR between slope classes is relatively small ($0.5 < FR < 1.5$). This suggests that, in this analysis, slope exerted only a limited influence on landslide susceptibility in the study area. In addition, this pattern may partly be influenced by the characteristics of the ITALICA landslide database. As noted by the database authors, landslides that took place in remote or uninhabited regions, or for which no institutional or media reports exist, are rarely included. Consequently, the inventory likely overrepresents events on gentler slopes and in lower-elevation areas near infrastructure, while underrepresenting those in steep, remote terrain. Finally, the ML framework is trained with the events mapped on a window of 3x3 pixels to compensate for the uncertainty in the co-ordinates and exact location of the recorded event. In hindsight, this window size may have extrapolated landslide occurrences to flatter terrains.

Regarding aspect, both flash floods and landslides are most likely to occur in areas facing South, and highly unlikely on flat surfaces. The orographic precipitation resulting from the Ligurian Sea partly explains the heavier rainfalls on South facing slopes contributing to an increase in both hazards.



Convex surfaces such as hilltops and ridges are very unlikely (FR<0.5) for flash floods to occur and unlikely (FR<1) for landslides to occur. On the contrary, flat areas were associated with the highest FR for flash floods, also observed in Chowdhury (2024).

For areas close to roads (0-22m) the FR is high (>1.5) for both flash floods and landslides and drastically drops for the larger distances. This agrees with (Mancini et al., 2010), who obtained the highest values of FR for landslides near the roads in Daunia, Italy located in the Apennines but also highlighted the bias of inventories.

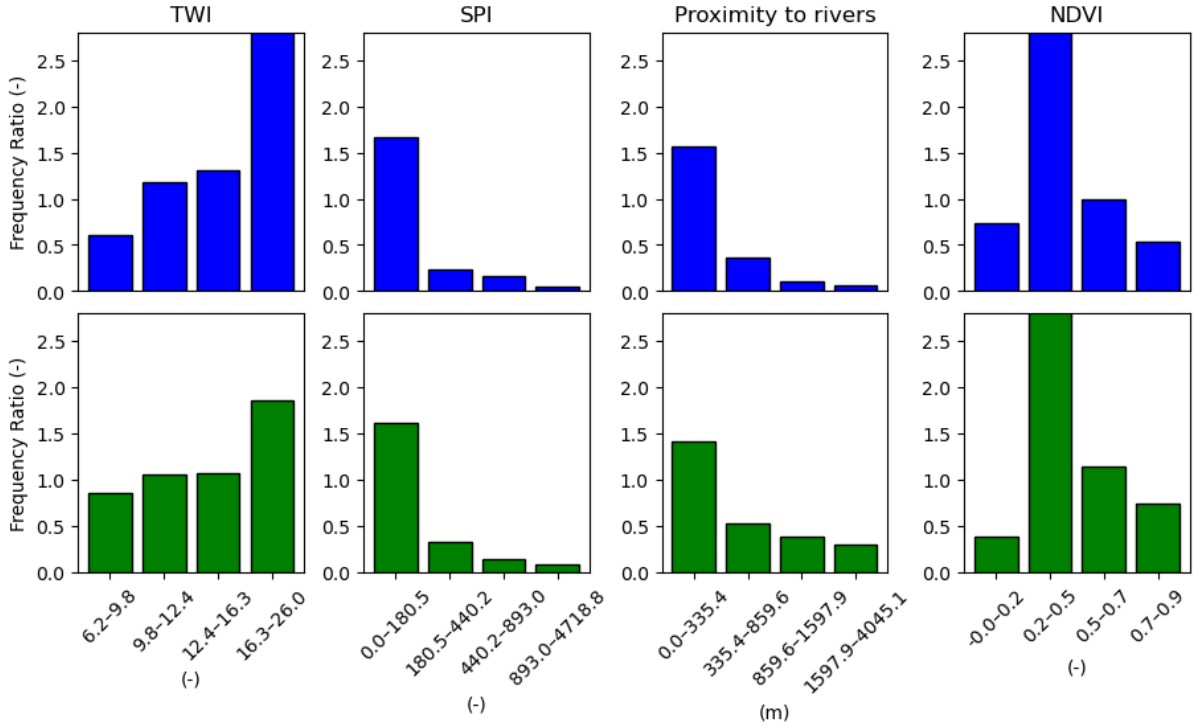

**Figure 8: Frequency ratio (FR) for flash floods (top) and landslides (bottom) for numerical variables continued.**

With respect to the Topographic Wetness Index (TWI), the highest FR values - which indicate areas that are most susceptible to saturation – correspond to the highest likelihood for both hazards (see Figure 8). Similar results were reported by (Chowdhury, 2024; Elghouat et al., 2024; Rayamajhi et al., 2025) who also found that areas with high TWI values are most prone to flooding. The FR decreases with decreasing TWI, a trend pronounced more for flash floods than for landslides.

On the contrary, areas with low Stream Power Index (SPI) – indicating lower erosive power of streams - exhibit the highest
FR (~1.6) for both hazards, with FR strongly decreasing as SPI increases. Similarly, Chowdhury (2024) obtained the highest FR values for SPI values close to 0.

Proximity to rivers showed a clear spatial relationship: the closer to rivers, the higher the likelihood of both flood and landslides. This likelihood decreases sharply with distance. Similar patterns have been reported for flash floods and floods in other studies (Elghouat et al., 2024; Pham et al., 2021; Rayamajhi et al., 2025).





In line with expectations, we obtain high FR values for areas with low NDVI (0.2-0.5), corresponding to bare soil or areas with limited vegetation which usually experience more frequent landslide and flash floods occurrences. With increasing NDVI values, we observe decreasing FR values.

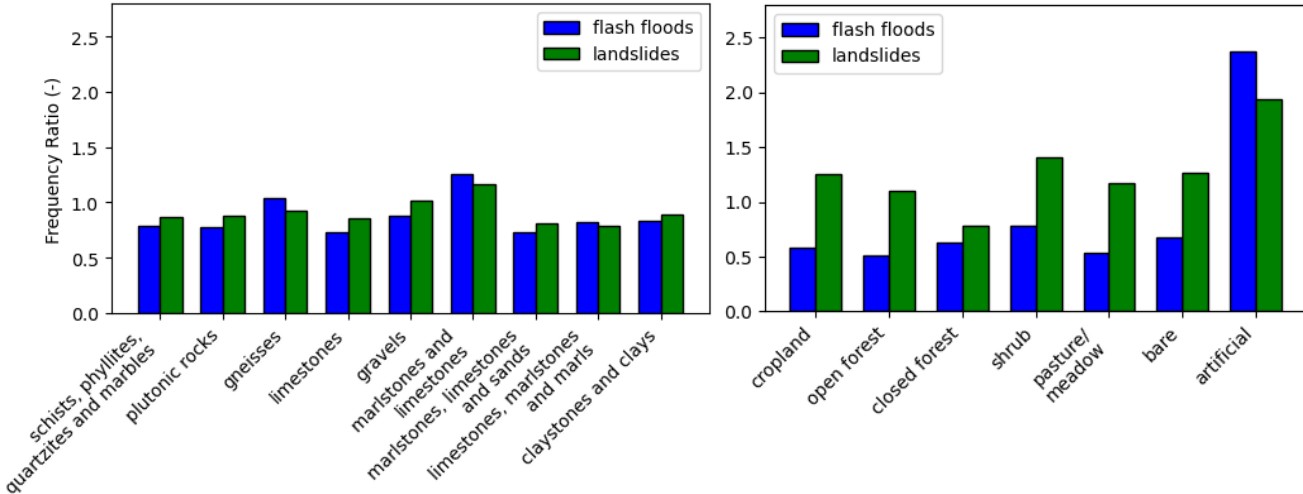

**Figure 9: Frequency ratio (FR) for sub surface lithology (left) and land cover (right).**

The influence of surface lithology and land cover on hazard occurrence is shown in Figure 9. For lithology, the highest FR values for both hazards correspond to marlstones and limestones which have low permeability. However, the FR for landslides and flash floods remained close to 1 (0.8-1.2 and 0.7-1.3 respectively) across all lithological classes, indicating that lithology exerted only a limited influence on landslide and flash flood occurrence in the study area.

Regarding land cover (Figure 9), the highest FR values are found for artificial soils. For flash floods this can be explained by
the limited infiltration capacity whereas for landslides, we expect the human-made changes to topography and loss of vegetation in urban areas to play a bigger role. Indeed, urbanization increases landslide hazard (Johnston et al., 2021). Flash floods are unlikely to occur in the other land cover classes (FR<1) whereas landslide likelihood occurrence is more evenly distributed across land cover classes except for closed forest in which they are unlikely to occur (FR<1). The latter confirms that strong root cohesion and vegetation cover enhance slope stability.






## 4.3 Hazard susceptibility maps

**Figure 10: Flash flood (left) and landslide (right) susceptibility maps for the three ML algorithms (LR - top, RF - middle, SVM - bottom).**


Figure 10 shows the susceptibility maps for flash floods and landslides for each of the ML models: Logistic Regression (LR), Random Forest (RF), and Support Vector Machine (SVM). The susceptibility maps obtained are continuous values ranging from 0 to 1 and have been reclassified into five discrete classes to improve interpretability of the outcomes using Jenks natural break (see Table A1 in the Appendix).

Overall, the maps follow a similar pattern for all algorithms and both hazard types. Along the coastline, areas of high susceptibility are widespread. Numerous villages and cities are located along this coast, including the historic city of Genoa, which has been repeatedly affected by severe flash floods in its urban valleys and by rainfall-induced landslides on the surrounding hills. Susceptibility is also high for both hazards along the Polcevera river valley, north of Genoa, where extensive



urbanization has led to the confinement or disappearance of natural waterways, ephemeral streams, and artificial channels,
thereby increasing the likelihood and severity of catastrophic events (Faccini et al., 2015; Lanza, 2003).

Although there are many areas susceptible to both hazards, pronounced differences between flash floods and landslides occur in the Western part of the Liguria region. Here the area with high landslide susceptibility is larger than the area with high flash flood susceptibility. This sector is characterized by relatively steep slopes and extensive crop land, including vineyards. In case of non-permanent crops and frequent tillage, soil stability tends to be reduced (Giarola et al., 2024) leading to high
occurrence of landslide events. In addition, the western Liguria region is driest part of the study area, with less than 600 mm / year, therefore leading to fewer flash floods recorded in the inventory.

Across all methods and ML algorithms, the total area identified as highly susceptible is larger for landslides than for flash floods. This difference largely reflects the characteristics of the two inventories (see Figure 1). The landslide events are more widespread, particularly in the northern and western sector of Liguria, whereas flash flood events are concentrated in the lower
valleys. It should be noted that, as mentioned in Sect. 0, a bias may be present. The landslide inventory used in the study consisted of 2416 events, whereas we only have 538 historic flash flood events.

## 4.4 Model Performance and Discriminatory Power

The framework was evaluated on the testing dataset. Among the three machine learning algorithms, the Random Forest (RF) approach achieved the highest accuracy (Table 2) and largest area under the curve (AUC) for both hazard types (Figure 11).
However, the susceptibility maps revealed that the RF method is relatively conservative, identifying a smaller area as highly susceptible (red zone) compared to Logistic Regression (LR) and Support Vector Machine (SVM) models. This suggests that while RF offers superior classification performance, it tends to minimize false positives by restricting the extent of predicted high-susceptibility zones.

These findings are consistent with other studies that reported Random Forest as the most effective model for predicting the
occurrence of flash floods and floods (e.g., Elghouat et al., 2024; Khodaei et al., 2025) and landslides (e.g., Youssef & Pourghasemi, 2021).





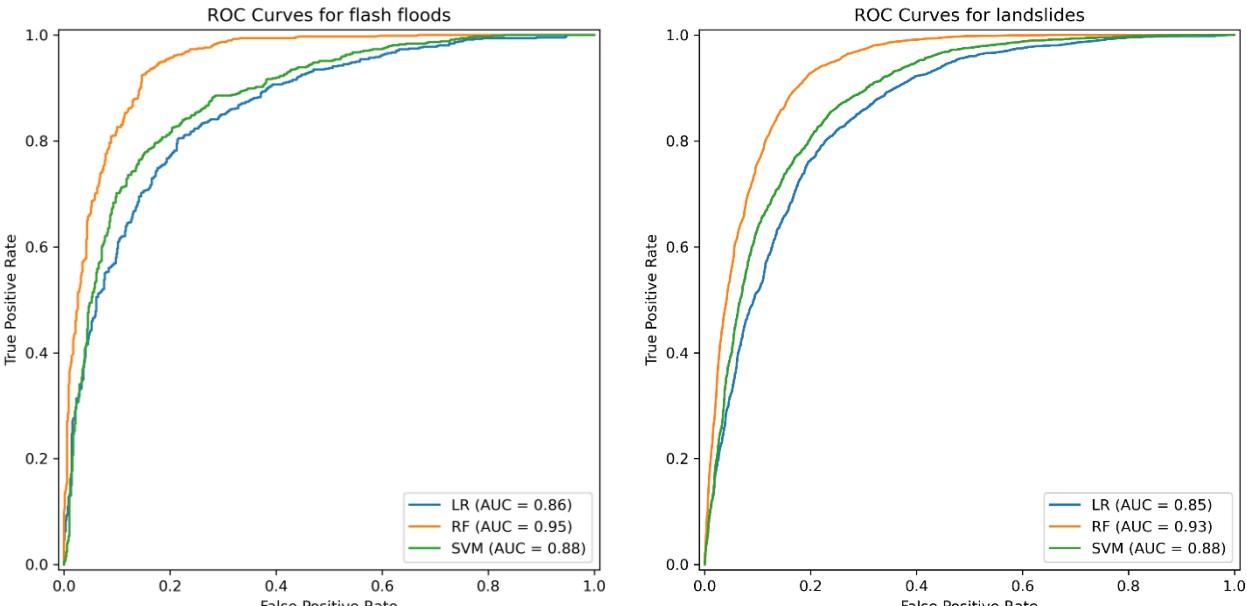

**Figure 11: Receiver operating characteristics (ROC) curves and the corresponding Area Under the Curve (AUC) for flash floods (left) and landslides (right) on the testing dataset for the three ML algorithms.**

**Table 2: Confusion matrix and accuracy for each ML model derived for the testing dataset.**

| Flash flood | | | |
|---|---|---|---|
| | **LR** | **RF** | **SVM** |
| **TP** | 537 | 593 | 544 |
| **FP** | 147 | 96 | 131 |
| **TN** | 540 | 591 | 556 |
| **FN** | 136 | 80 | 129 |
| **Accuracy** | 0.79 | 0.87 | 0.81 |
| Landslide | | | |
| | **LR** | **RF** | **SVM** |
| **TP** | 3091 | 3331 | 3209 |
| **FP** | 1027 | 665 | 976 |
| **TN** | 2666 | 3028 | 2717 |
| **FN** | 583 | 343 | 465 |
| **Accuracy** | 0.78 | 0.86 | 0.80 |





## 5 Discussion

The results obtained in this study provide new insights into the spatial patterns and drivers of both flood and landslide hazards
in Liguria. In the following discussion, we analyse the performance of the adopted models, the spatial correspondence between
flash floods and landslides, and the comparison with regional and continental susceptibility maps. We then address the
influence of data completeness and model assumptions, before outlining the main limitations and implications for multi-hazard
risk assessment.

### 5.1 Model performance and framework consistency

The unified framework applied in this study ensured methodological consistency between the analyses of flash floods and
landslides, allowing the same set of conditioning factors and modelling procedures to be used for both hazards. All tested
models showed good predictive capability, with Random Forest achieving the best overall performance (AUC=0.95 and
accuracy=87% for flash floods, and AUC=0.93 and accuracy=86% for landslides). The results confirm that the adopted set of
conditioning factors can effectively describe both processes. This suggests that the same modelling structure can be
successfully applied to different types of hazards, allowing a homogeneous multi-hazard evaluation across the regional scale.

### 5.2 Combined hazard susceptibility analysis

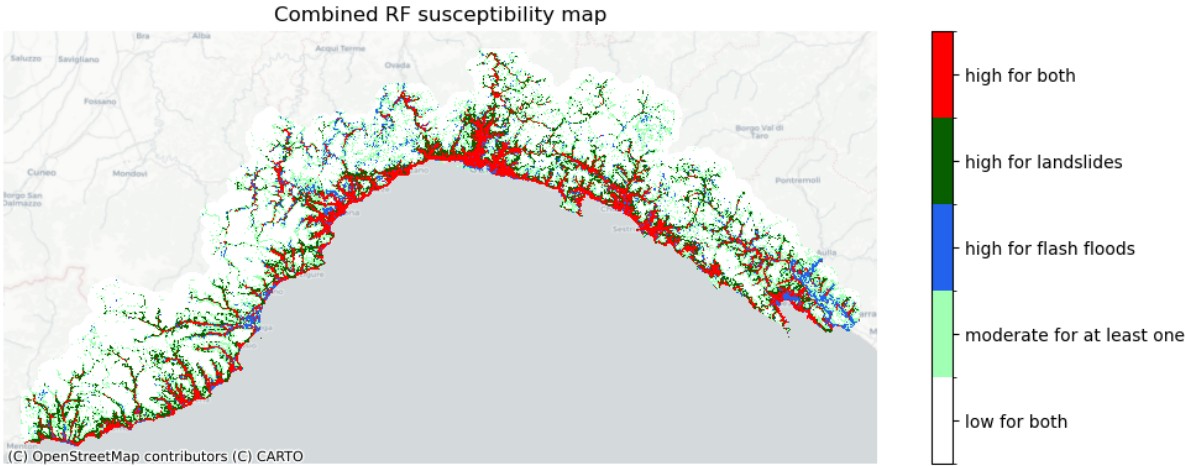

**Figure 12: Combined RF susceptibility map.**

We overlaid the flash flood and landslide RF susceptibility maps into one as shown in Figure 12 to identify areas susceptible
to both. The classification used is found in Table  of the Appendix. There is a large overlap in terms of high susceptibility for
both hazards mainly along the coast but also further inland along valleys (see DEM map in Figure 3). The areas where the
susceptibility is high for landslides but not for flash floods (dark green) is more spread out across Liguria as opposed to vice



versa where the blue areas are mostly clustered in three areas (from left to right): (i) in the city of Albenga along the Centa
River, (ii) close to Savona inland between two valleys, and (iii) along the Magra and Vara rivers including the surrounding
urban areas (e.g. La Spezia). The higher precipitation received on the East (Sect. 2 Study Area) and the higher relative historical
flash flood occurrences on the East (Figure 1) can help explain the larger spatial extent of the rightmost blue cluster.

Nevertheless, the spatial susceptibility patterns for both hazards are mostly similar which is crucial to highlight as one hazard
may amplify the other (Borga et al., 2014), e.g. a landslide may block a stream elevating the flood hazard. Gill & Malamud
(2014) found that floods and landslides both trigger and increase each other's probability, hence, the importance of studying
them together.

### 5.3 Spatial patterns and hazard interconnections

The susceptibility maps show a marked spatial correspondence between flash floods and landslides, mainly along the coastal
areas and the main valleys of Liguria. These zones are characterized by steep slopes, low-permeability lithologies and strong
anthropogenic pressure, which favour the simultaneous occurrence of both processes. Flash floods tend to be concentrated in
low-elevation and urbanized valleys, whereas landslides extend toward the upper slopes and inland sectors. The two processes,
although controlled by partly different conditions, share several predisposing factors. This confirms that they should be
analysed within an integrated and unified framework. Our methodology, however, does not consider joined occurrence of flash
floods and landslides as in e.g. Claassen et al., (2023) for which multi-hazard events are defined as events overlapping spatially
with a time lag between their dates of occurrence. Due to the limited spatial scale of flash floods and landslides we expect a
limited overlap but a more realistic event footprint could be studied such as slope units for landslides (Woodard et al., 2024)
and sub catchments for flash floods (Yin et al., 2023).

### 5.4 Comparison with European landslide susceptibility map

The generated Random Forest landslide susceptibility map was compared against the European Landslide Susceptibility Map
version 2 (ELSUS v2) (Joint Research Centre (JRC), European Commission, 2018; Wilde et al., 2018). The ELSUS map was
developed using elevation, climatic conditions, slope, sub-surface lithology, and land cover datasets. Compared to European
regions, the geohydrological vulnerability of Liguria is particularly high (Faccini et al., 2015). This is evident from the ELSUS
map, in which most of the region is classified as having high or very high susceptibility. The Western section nearly completely
has a high susceptibility. For the coastline, landslide susceptibility is lower than in the here generated maps. Possibly, for the
ELSUS map more emphasis was paid to the slope and elevation that are both lower in these areas, leading to a lower landslide
susceptibility.

Nevertheless, because the ELSUS dataset provides continental-scale classes, the spatial granularity for small regions like
Liguria is limited. Although our study also uses a globally applicable framework, by focusing on a specific region, a more
detailed and locally representative susceptibility map could be generated.



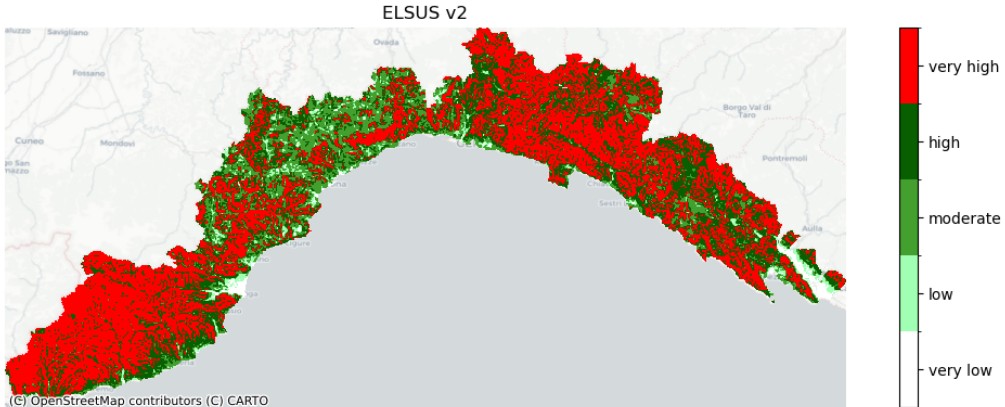

**Figure 13: Landslide susceptibility map for Liguria according to the European Landslide Susceptibility Map version 2 (ELSUS v2; EC, 2018)**

## 5.5 Role of inventories and data completeness

The ML framework is globally applicable; however, the accuracy of generated susceptibility maps remains highly dependent on the completeness of hazard inventories. In the Liguria region, the frequency of occurrence of events is relatively high, and records are generally well documented. Moreover, the inventory prepared for this study relied on multiple sources to increase the coverage in space and time partially minimizing the spatial bias (Bornaetxea et al., 2023). Nevertheless, biases remain in the spatial distribution of recorded events with more observations near roads and in lower-elevation areas, reflecting the greater accessibility and population density in these zones. As a consequence, the susceptibilities are potentially less reliable for remote areas. The influence of the spatial bias could be explored by removing part of the inventory contributing to it and assessing the difference in susceptibility outcomes (Steger et al., 2017).

Previous studies (Free et al., 2022; Modrick and Georgakakos, 2015; Uwihirwe et al., 2022) have shown that in many other, especially remote, regions of the world the number of recorded events is substantially smaller. Global inventories can fill some gaps but are often coarse, incomplete or inconsistent in spatial and temporal coverage. This highlights the added value of extending the national inventories (AVI and ITALICA) with a more local data (ARPAL), supported by the use of Large Language Models to enable automatic information extraction. This approach worked for Liguria, but relied on the available institutional event reports. In more data-sparse regions, collecting event data would be more difficult, but LLMs could potentially be applied for mining newspapers, social media and global disaster report archives.

The exact definition of flash floods influences the number and specifications of the events considered in the final inventory and hence the susceptibility maps. We note that our definition of flash floods for the AVI database is more restrictive than that of Vennari et al., (2016) which focused on Campania (to the South but also extending from the Sea to the Apennine) where they assumed all floods in the AVI inventory to be flash floods except the ones in alluvial plains. In our case, we explicitly



used the description and the precipitation duration (if available) to select flash flood events, consequently many entries were
excluded.

**5.6 Limitations and implications**

Although the proposed approach produced promising results, several limitations should be acknowledged.  The current
framework relies on static conditioning factors and does not yet incorporate dynamic variables such as rainfall, soil moisture,
or land-use change. Including these parameters would enhance the capacity of the models to capture transient conditions and
to update susceptibility in near real time. Moreover, it holds the potential to be applied in early warning mode using forecasted
precipitation and soil moisture conditions as dynamic inputs, by adding information on the local hazard susceptibility, as we
previously explored in Uwihirwe et al. (2022). Applying the framework for longer-term climate projections will be more
complex as, next to the climate change induced rainfall changes, the landscape and vegetation will also be influenced by
anthropogenic forces which are even more uncertain. This could potentially make soils more erodible, increase the
impermeable surface or even further reduce the network of natural waterways (Stalhandske et al., 2024) and thus require
changes in the static maps underlying the framework. Despite these limitations, the results demonstrate the potential of the
proposed framework for regional multi-hazard susceptibility assessment.

**6 Conclusion**

In this study a globally applicable Machine Learning (ML) Framework for landslide and flash flood susceptibility mapping
was applied and evaluated in the Italian region Liguria. The Framework supports multi-hazard analysis by enabling a one-to-
one comparison indicating the areas most susceptible to either one or both hazard types. The resulting maps can help to identify
where intervention or adaptation is most needed.
The framework was tested for Liguria, located in north-western Italy, which provided an ideal case study due to its high
susceptibility to both flash floods and landslides. The application of the ML framework resulted in accuracies of 0.87 and 0.86
for flash flood and landslide prediction, respectively. The general pattern in the resulting susceptibility maps is comparable,
although flash floods most often occur in river valleys and urban areas whereas landslide susceptibility is also high in the upper
courses of river catchments along ephemeral streams. In Liguria, very high susceptibility occurs along the coastline, where
many villages and cities are located. The proposed framework can be applied in any region of the world, although the accuracy
remains dependent on the completeness and quality of local event inventories. This underlines the importance of continuously
improving the documentation and reporting of hazardous events to support reliable and consistent multi-hazard assessments.
Beyond its application in Liguria, the unified framework provides a practical foundation for regional-scale risk management.
Its ability to integrate different hazard types under a consistent modelling structure enhances the comparability of results and



supports the development of integrated mitigation strategies. Future work should focus on incorporating dynamic variables
such as rainfall change, as well as evaluating model performance under changing climate conditions.

## Data availability

The input data including the inventories and the code used to create the susceptibility maps in this study is available at Zenodo: https://doi.org/10.5281/zenodo.17579993 (fork of https://github.com/openearth/lhat/tree/main).

## Author Contribution

AR: conceptualization, data curation, formal analysis, visualization and writing.

CG: conceptualization and data curation.

MM: funding acquisition, supervision, investigation, methodology and writing.

FSW: conceptualization, funding acquisition, supervision, methodology and writing.

## Competing interests

The authors declare that they have no conflict of interest.

## Acknowledgement

This research has received funding from the European Union's Horizon Europe—the Framework Programme for Research and Innovation (MEDiate (Grant No. 101074075)). The authors thank Jing Deng for running the LLM that enabled the
extension of the flash flood and landslide inventories.

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





**Appendix A**

**Table A1: Jenks breaks used for the flash flood and landslide susceptibility maps for the three ML algorithms.**

| Flash flood | | | |
|---|---|---|---|
| | **LR** | **RF** | **SVM** |
| **Very low** | 0.0 – 0.131 | 0.0 – 0.137 | 0.0 – 0.124 |
| **Low** | 0.131 – 0.295 | 0.137 – 0.275 | 0.124 – 0.265 |
| **Moderate** | 0.295 – 0.478 | 0.275 – 0.451 | 0.265 – 0.470 |
| **High** | 0.478 – 0.697 | 0.451 – 0.681 | 0.470 – 0.707 |
| **Very high** | 0.697 – 1.0 | 0.681 – 1.0 | 0.707 – 1.0 |
| Landslide | | | |
| | **LR** | **RF** | **SVM** |
| **Very low** | 0.0 – 0.119 | 0.0 – 0.120 | 0.0 – 0.121 |
| **Low** | 0.119 – 0.295 | 0.120 – 0.280 | 0.121-0.289 |
| **Moderate** | 0.295 – 0.485 | 0.280 – 0.475 | 0.289 – 0.495 |
| **High** | 0.485 – 0.679 | 0.475 – 0.695 | 0.495 – 0.704 |
| **Very high** | 0.679 – 1.0 | 0.695 – 1.0 | 0.704 – 1.0 |

640

**Table A2: Classification used in the overlay of the RF flash flood and landslide susceptibility map.**

| | | Flash Flood | | | | |
|---|---|---|---|---|---|---|
| | | **Very High** | **High** | **Moderate** | **Low** | **Very Low** |
| **Landslide** | **Very High** | 🟥 | 🟥 | 🟩 | 🟩 | 🟩 |
| | **High** | 🟥 | 🟥 | 🟩 | 🟩 | 🟩 |
| | **Moderate** | 🟦 | 🟦 | 🟩 | 🟩 | 🟩 |
| | **Low** | 🟦 | 🟦 | 🟩 | ⬜ | ⬜ |
| | **Very Low** | 🟦 | 🟦 | 🟩 | ⬜ | ⬜ |