# Peer review of "A single framework for assessing flash flood and landslide susceptibility: an application to the Mediterranean"

_EGUsphere, 2025_

## Referee Comment (RC1)

The study presents and evaluates a globally applicable machine-learning framework for integrated landslide and flash flood susceptibility mapping, demonstrated in the multi-hazard-prone Liguria region of Italy, with the aim of supporting effective risk management. The topic is relevant and promising; However, several parts of the manuscript require further modifications and improvement before it can be considered for publication in the journal

**Title:** I understand that the study area is well presented within the Mediterranean context. However, a minor change to the title would improve clarity and better specify the case study. Consider replacing "Mediterranean" with "Liguria region, Italy."

Additional comments and suggestions are outlined below:

**Introduction Section:**

**Line 41:** The distinction between land cover and vegetation, is unclear. For instance, is vegetation considered part of land cover, or are you referring to a vegetation index such as NDVI?

**Lines 46–47:** "...due to high susceptibility...landslides." How is this demonstrated? Please cite at least one study to support this statement and make it more precise.

**Study Area:** Please provide information on recorded damage and economic losses in the region, if available.

**Line 61:** "...period of 1910–2010...1990." Is more recent meteorological data available, or is it excluded from the modeling? Also, is the inventory data collected for the same period, or how was it determined?

**Line 68:** Please rephrase, as the term "hydrology" is too general and should be more specific in this context.

**Line 69:** "...from 1 to 5 hours." Please provide a citation.

**Figure 1:** The map requires the following improvements:

- Add scale bars for both the left and right panels.
- Include a north arrow.
- The black lines and rectangle are somewhat confusing. I suggest either removing the two lines (currently only indicating flash floods) or adjusting them to also indicate landslide areas.
- Consider using the term "Elevation" instead of "DEM" in the legends for both flash flood and landslide maps. While it is technically a DEM, "Elevation" would be clearer for the reader.

- Ensure that the coordinates (longitude and latitude) are consistent across all maps.

**Data Section:**

**Line 84:** "(1) the AVI…Italia)" requires a proper citation and/or a link to the data source.

**Line 89:** "OpenStreetMap" needs to be cited.

**Line 90:** "(2) ARPAL" also requires a citation.

**Line 99:** Replace "didn't" with "did not."

**Figure 2:** Include the year(s) of both inventories in the figure caption.

**Lines 126–128:** "…coarse-resolution…finer resolution." Please clarify precisely what do you mean and provide details. This statement can also be supported with a reference.

**Table 1:**

- Land cover: Have you checked for newer, higher-resolution data (e.g., ~50 m) to better match your DEM and improve consistency? JRC has recently updated their datasets.
- Distance to roads: You mention "There is no…resolution," but it is possible to provide the resolution or scale of OpenStreetMap.

**Figure 3:** Apply the same suggestions as for Figure 1, and:

- Remove shadows from the background of each variable map, as they make the study area polygon misaligned with calculated areas. Alternatively, remove the borders.
- Adjust font sizes for clarity.

**Figure 4:** Apply the same improvements as above.

**Line 141:** Replace "DEM" with "elevation." While elevation is extracted from a DEM, the DEM itself is not a topographic factor, elevation is.

I also suggest overlaying the landslide and flash flood inventories on these maps to visually examine the factors prior to modeling.

**Methods Section (3.2):**

Please specify the versions of any software used (e.g., GIS, modeling tools).

**Lines 191–192:** "Only for…the results." Could you clarify this statement further, and explain how it was determined?

**Model Calibration:** Please provide more details on the calibration phase to enhance clarity and reproducibility. Include references or studies that guided your calibration approach.

**Results Section:** This section should be more comprehensive, with a clearer descriptive analysis of the findings. Currently, the maps are too small to allow independent interpretation; consider adding zoomed-in areas or inset maps.

Additionally, the distinction between the Results and Discussion sections should be strengthened. The Results should focus on presenting outcomes and facts, while interpretation and comparison with other studies should be reserved for the Discussion to improve the overall flow. For example, **Lines** 276–279, 295–297, and 300–306 should be rephrased or moved to the Discussion; please review the manuscript accordingly.

**Figures 7 and 8:** The threshold values of the factors (variables) are not optimally aligned with the desired thresholds and appear slightly shifted to the right. Redesigning their positioning and size is recommended to improve clarity and facilitate interpretation of the results.

**Line 307:** "proximity…relationship." Please specify the figure from which this information is derived (e.g., Figure 8) to improve clarity.

**Figure 10:** Apply the same suggestions as for Figure 1. I strongly recommend adding zoomed-in inset maps to better highlight key results. This will allow a clearer visual interpretation and more effective discussion of the findings, as relying solely on numerical assessments does not fully convey the spatial patterns of the study.

**Table 2:** Please provide more details on how the confusion matrix was generated. Are all numbers based on points or polygons? If polygons were used, an area-based accuracy assessment might be more appropriate, and showing the full spatial representation would be more informative than only the numerical summary.

**Suggestion:**

- The table could be presented more concisely by combining both flash flood and landslide into a single row. Including the percentage of each category alongside the numbers would provide a clearer picture of the proportions in addition to the overall accuracy.
- Including confusion matrix results for the training dataset, in addition to the test dataset, would also help readers better

understand the model's performance and allow for a more meaningful comparison between the datasets.

**Discussion:** Please discuss the transferability of this model (workflow), including any modifications that may be needed. Could you also clarify whether you would recommend its application at local scales and what improvements would be necessary for such use?

It would be helpful to provide a brief explanation regarding the confusion matrix in Table 2, here in the Discussion. The findings look promising, and adding this context would help readers better understand the practical significance of the observed differences.

**Line 269:** "The results...study" better to support this by providing table, map, figure, then continue.

**Figure 12:** First, please include information similar to the previous figures and use the maximum size allowed by the journal to improve readability. For Figure 12, the map could be presented to better support visual assessment of the results alongside the statistics. While the overall map is good, its usefulness is limited without zoomed-in views (inset maps) of key locations. If you overlay the inventories (which I recommend), consider showing them only on these zoomed-in subsets to avoid excessive density and maintain clarity, rather than overlaying everything on the main map.

**Lines 389–390:** "...in the city... (e.g., La Spezia)." These locations are difficult to identify on the map. Consider highlighting them on the existing map or preparing a separate figure showing all locations discussed in the text. This would visually strengthen the analysis and improve the clarity and quality of the results (see the previous comment on Figure 12).

**Line 399:** "areas and...Liguria" may be difficult to locate for readers unfamiliar with the study area. These should be highlighted on the map or provide a zoomed-in one, as suggested previously, to improve clarity and reader understanding.

**Line 417:** The reference to "ELSUS" should be properly cited.

**Figure 13:** Please add a scale and other map elements as before. Additionally, indicate the original scale at which ELSUS v2 was mapped.

**Lines 444-445:** "..,consequently...excludes." Have you documented this? Please provide the number of cases that were excluded.